# Influence of Heparan Sulfate Proteoglycans and Factor X on species D Human Adenovirus Uptake and Transduction

**DOI:** 10.3390/v15010055

**Published:** 2022-12-24

**Authors:** Katrin Schröer, Montaha Alshawabkeh, Sebastian Schellhorn, Katrin Bronder, Wenli Zhang, Anja Ehrhardt

**Affiliations:** Virology and Microbiology, Center for Biomedical Education and Research (ZBAF), Department of Human Medicine, Witten/Herdecke University, 58453 Witten, Germany

**Keywords:** Human Adenovirus Species D, Chinese Hamster Ovarian Cell Lines, heparan sulfate proteoglycan, human blood-coagulation sactor X

## Abstract

More than 100 human adenovirus (Ad) types were identified, of which species D comprises the largest group. Heparan sulfate proteoglycans (HSPGs) were shown to function as cell surface receptors for cell binding and uptake of some Ads, but a systematic analysis of species D Ads is lacking. Previous research focused on Ad5 and blood coagulation factor X (FX) complexes, which revealed that Ad5 can transduce cells with low expression levels of its main coxsackievirus-adenovirus receptor in the presence of high HSPG expression levels in a FX dependent manner. Based on our reporter gene-tagged Ad-library, we explored for the first time a broad spectrum of species D Ads to study the role of HSPG on their cellular uptake. This study was performed on three Chinese Hamster Ovary (CHO) cell lines with different forms of HSPG (only proteoglycan (745), non-sulfated HSPG (606) or sulfated HSPG (K1)). The effect of Ad:FX complexes on Ad uptake was explored in the presence of physiological levels of FX in blood (6–10 µg/mL). We found that sulfation of HSPG plays an important role in cellular uptake and transduction of FX-bound Ad5 but neither HSPG nor FX influenced uptake of all tested species D Ads. Because FX has no influence on transduction efficiencies of species D Ads and therefore may not bind to them, these Ads may not be protected from attack by neutralizing IgM antibodies or the complement pathway, which may have implications for species D Ads used as vaccine and gene therapy vectors.

## 1. Introduction

Human adenoviruses (Ads) are non-enveloped viruses with an icosahedral shaped capsid and which have a linear double-stranded DNA genome of about 36 kbp. The capsid comprises 240 hexon bases, which build the 20 faces of the capsid, and 12 penton bases at each vertice with protruding fiber and knob [1,2,3,4]. To date, 113 Ad types subdivided into species A to G are known [5] of which species D represents the largest group [5,6]. Clinically, species D Ads cause eye infections such as conjunctivitis and some species D Ad types are associated with severe epidemic keratoconjunctivitis (EKC) [7]. Species D Ads, and in particular human Ad type 26, are used as vaccine vectors, e.g., for prevention of infection with Ebola and Zika virus and recently for prevention of SARS-CoV-2 infections in the COVID-19 pandemic [8]. Current research has shown that some Ads of species B, C and F (type 3, -5, -35, -40 and -41) utilize Heparan sulfate proteoglycans (HSPGs) as receptors for binding and uptake into cells [9,10,11], but a systematic analysis of HSPG binding properties of species D Ads is lacking. HSPGs are ubiquitously expressed in almost all mammalian cells and comprise a core protein out of proteoglycan (PG) with one or more covalently linked unbranched heparan sulfate chains (HS) [12,13,14,15]. HSPGs are located either on plasma membranes, on secreted extracellular matrix, or on secretory vesicles. The membrane-located HSPGs are interacting with integrins and some other cell adhesion receptors to promote cell–cell communication and motility [12,15]. They act as the main receptor for proteases and can interact with chemokines, cytokines, growth factors and morphogens to protect the cell against proteolysis [12,13,15]. The HS are polyanionic, therefore, they are highly negatively charged and consist of 40 to 300 sugar residues, sulfate groups and uronic acids [9,12,14]. They can reach a size between 20 and 150 nm, which creates an enormous polydispersity [12]. The sulfation pattern of HS differs depending on the cell type [12,16], wherein liver HS have the highest levels of sulfation among tissues [9,14].

Because of the strong negative charge, HS can interact electrostatically with the viral surface of enveloped viruses or the capsid of non-enveloped viruses [15]. Therefore, they are suggested to act as cell binding partners and/or receptors for human immunodeficiency viruses (HIV) [9,15], human papillomaviruses (HPV) [9,15], adeno-associated viruses (AAV) [13], herpes simplex virus (HSV) [9,13,15], dengue virus [15] and some adenoviruses [9,10,11]. For adenovirus, previous research on adenovirus type 5 (Ad5) and blood coagulation factor X (FX) revealed that Ad5 can transduce cells with low expression levels of the main Ad5-coxsackievirus-adenovirus receptor (CAR), but with high levels of HSPG expression in a FX dependent manner [1,9,13,14,17,18,19,20,21,22]. It was also shown, that the interaction of FX with HSPG is dependent on the sulfation pattern of the heparan chains [9].

Physiological concentrations of FX in the blood plasma range from of 6 to 10 µg/mL [23,24]. It is a calcium- and vitamin K-dependent proenzyme of a serine protease containing epidermal growth factor 1 and 2 (EGF1 & 2), and a Gla (γ-carboxylated glutamic acid) domain [19]. Electron cryomicroscopy showed that the Gla domain of FX binds to the cavity of a hexon trimer on the Ad5 capsid [18], resulting in 205 FX molecules bound per viral particle [19], thus protecting the Ads from neutralization by human sera via IgM and classical complement pathway [14,17,25]. The interaction between Ad5 and FX occurs between hypervariable loops 5 and 7 of the Ad hexon and a lysine residue from FX in the Gla domain. The serine protease domain of FX binds to the HSPG on the host cells [14,17]. A schematic overview of the bridging of FX between Ad5 capsid and HSPG is shown in Figure 1. Additionally, α_v_ integrins are needed as secondary receptors for a successful uptake of Ad5:FX-complex via HSPG into the host cell [1,22].

Because of the potency of species D Ads as vaccines and for gene therapy, virus–host cell receptor interactions need to be explored in further detail. There is some information about the receptor usage of single species D Ad types in the literature [8,26], e.g., Ad8, 19a, 6 and 37 use sialic acid (SA) as primary receptor [27,28,29] and Ad9, 10 and 24 use CAR [30,31], but in-depth analyzes of a broader spectrum of species D Ads, especially for utilizing HSPGs as a receptor are lacking. Therefore, the focus of this work was investigating the interaction between HSPGs, blood-coagulation factor X (FX) and the Ad itself.

## 2. Materials and Methods

### 2.1. Cell Culture

The Chinese hamster ovarian cell lines CHO-K1 [32], CHO-pgsE-606 [33] and CHO-pgsA-745 [34] were obtained from André Lieber from the University of Washington. All used cell lines were cultured in Dulbecco’s modified Eagle’s medium (DMEM; Pan Biotech, Aidenbach, Germany) supplemented with 10% fetal bovine serum (FBS Gold; Pan Biotech, Aidenbach, Germany), 1% non-essential amino acids (MEM NEAA; Pan Biotech, Aidenbach, Germany) and 1% penicillin streptomycin (Pan Biotech, Aidenbach, Germany) in an incubator at 37 °C and 5% CO_2_.

### 2.2. Adenovirus Library

All used Ads, including HAdV-C5 as control and Ad types HAdV-E4, HAdV-B3, -14, -16, -21, -35, -50 and HAdV-D9, -10, -17, -20, -24, -26, -27, -33, -37, -69, -70, -73 and -74 with a GLN-cassette were previously generated [35]. The GLN-cassette encodes the multicistronic TurboGFP (green fluorescent protein (G)), NanoLuc luciferase (L) and a neomycin/kanamycin selection marker (N)) cloned into the non-essential early region E3. Therefore, the Ads can be detected by luciferase assays or flow cytometric GFP assays.

### 2.3. Flow Cytometric Analysis of HSPG

For identifying HSPG on the used CHO cell lines, the cells were detached from the cell culture plate by scraping, to keep the extracellular HSPG intact. Cells were then counted, aliquoted to 300,000 cells, washed twice with DPBS (Pan Biotech, Aidenbach, Germany) and resuspended in 100 µL of DPBS. The first antibody, mouse anti-human heparan sulfate IgM (10E4 epitope, clone 8.S.087 [9,20]; N1130, Biomol, Hamburg, Germany), was added at a dilution of 1:100, and the cells were then incubated in a thermo block for 30 min at 37 °C with low-speed shaking at 350 rpm. After incubation, the cells were washed again twice with DPBS and resuspended in 100 µL of DPBS. Secondary goat anti-mouse-IgM antibody with AlexaFluor® 488 (ab150121, Abcam, Boston, MA USA) was added at a dilution of 1:1000 and also incubated in a thermo block for 30 min at 37 °C with low-speed shaking at 350 rpm. After incubation, the cells were washed twice with DPBS again and resuspended in 120 µL DPBS, then transferred to a 96-well micro test plate (Sarstedt, 82.1581) for FACS measurement (CytoFLEX from Beckman Coulter, Brea, CA, USA; CytExpert software version 2.4). For further details of the analysis see Appendix A.

### 2.4. Virus Transduction Detected by Luciferase Assay

Because all used Ads contain a GLN-cassette including the gene for NanoLuc luciferase, they can be detected using NanoGlo® Luciferase Assay System (Promega N1130, Walldorf, Germany). CHO cells were prepared with 100,000 cells/mL, seeded in a 96-well plate for 24 h and infected with different virus particles per cell (vpc). At 24 h after infection, the luciferase assay was performed by adding 1:1 luciferase buffer and substrate mix to each well. After 5 min incubation, the cells were transferred to a non-treated Nunc® MicroWell™ 96-well plate (Thermo Scientific, Waltham, MA, USA) and luminescence was measured in a plate reader (Infinite F Plex Tecan, Maennedorf, Switzerland).

For testing species D Ad uptake with and without the physiological levels of FX, the same experimental setup was used as described above except adding simultaneously 6, 8 or 10 µg/mL FX (Human Factor X, Haematologic Technologies, Essex Junction, VT, USA) in Opti MEM® (OMEM, Thermo Scientific, MA, USA) during infection with the Ads. OMEM was used because it is a serum-reduced media to avoid additional serum interference on the effect of FX. As a positive control for the impact of FX, HAdV5 was used.

### 2.5. Virus Cellular Entry Detected by Duplex qPCR

To compare the virus entry into all cell lines by preincubating with or without FX, a duplex qPCR was performed. Previously, all cell lines were seeded in a 24 well plate with 200,000 cells/well for 24 h. The medium was changed to Opti-MEM® (OMEM) and was infected with 1000 vpc with or without FX in a concentration of 10 µg/mL. After 3 h incubation, cells were collected, and the total genomic DNA (gDNA) was isolated with the Monarch Genomic DNA Purification Kit (T3010L NEB, Frankfurt am Main, Germany). For duplex qPCR, as the house-keeping gene for the CHO cell lines, glyceraldehyde 3-phosphate dehydrogenase (GAPDH; HEX) was used. Ads were detected by targeting the neomycin resistance gene on the GLN-cassette (FAM). As polymerase, the Takyon™ No Rox Probe MasterMix dTTP (UF-NPMT-B0701, Takyon, Seraing, Beligium) was used. The primers were used in an overall concentration of 300 nM and the probe of 200 nM (Table 1).

### 2.6. Statistics

All shown data are displayed as mean ± SEM. Statistical comparison was made using the two-tailed t test. A value of *p* ≤ 0.05 (*) was considered as a significant difference to the respective control group, *p* ≤ 0.01 (**) very significant and *p* ≤ 0.001 (***) highly significant.

## 3. Results

### 3.1. HSPG-Expression of the Used CHO Cell Lines

To prove the different HSPG depletions of the three used CHO cell lines, flow cytometric analysis was performed. In Figure 2A, the different HSPG-depletions of the CHO cell lines are displayed. CHO-K1 cells contain the wild type version of HSPG: a membrane-bound proteoglycan with protruding sulfated heparan chains [10,13]. The CHO-606 cells express non-sulfated heparan chains and CHO-745 retains the proteoglycan [10,13]. Cells were analyzed by flow cytometry and we found that 99% of CHO-K1 cells were positive for HSPG, and up to 12% sulfated heparan chains could be detected for the CHO-745 and -606 cells (Figure 2B, cyan).

### 3.2. Sulfation of Heparan Chains Is Relevant for Adenovirus Transduction via HSPG

To determine the transduction efficiency of our GLN-encoding Ads, the three different CHO cell lines were infected with a range of 1000 to 10,000 vpc for each virus. At 24 hours post infection (hpi), luciferase assays were performed to compare the transduction rates of all used Ads among the HSPG-depleted CHO cells. The results displayed in Figure 3 show the transduction efficiency of CHO-745 and CHO-606 cells normalized to CHO-K1 cells for 5000 vpc and are subdivided into species B (Figure 3A), C, E (Figure 3B), and D derived Ads (Figure 3C). The results for the other vpc can be found in Appendix A. In Table 2, the results of the GLN-Ad library screening are summarized by species and highest transduction efficiencies in the respective CHO cell lines. Additionally, the receptor and adapter usage of the different Ad species are displayed [36,37,38].

In the non-sulfated HSPG-expressing CHO cell line 606, species C and E showed a significant decrease in transduction compared to the wild type HSPG (Figure 3B). All the tested species B Ads in CHO-606 showed a decrease in transduction efficiency or a similar one, whereby only the decrease from Ad14 and 21 was significant (Figure 3A). From 13 tested species D Ads, 10 revealed lower or similar transduction efficiency in CHO-606 compared to the wild type version of HSPG in CHO-K1 cells. Only Ad10, 20 and 24 showed a small increasing effect of 5-, 1.5- and 2-fold (Figure 3C). Taken together, these results show that the sulfation status of the heparan chains plays an important role in successful Ad transduction of any tested species with only a few of species D Ads, which showed a slightly enhanced transduction efficiency in sulfation-ablated HSPG-expressing cells.

Furthermore, for all tested species B Ads, except Ad3, and for the used species C and E Ads, significantly lower or similar transduction efficiency can be observed in the non-HSPG-expressing CHO cell line 745 compared to wild type HSPG-expressing CHO cell line K1 (Figure 3A,B). Ten out of 13 tested species D Ads showed a lower or similar transduction efficiency in completely HS-ablated cells CHO-745 compared to the wild type form of HSPG in CHO-K1 cells. Only Ad10, -24 and -26 showed 19-, 4- and 2-fold enhanced transduction efficiencies, respectively (Figure 3C). The slightly increasing transduction effect from Ad20 in CHO-606 cells was reduced in CHO-745 cells. The Ad10 and 24 transduction efficiencies were enhanced in HSPG-ablated CHO-745 cells compared to sulfation-ablated CHO-606 cells. A reason for this may be the reduction of sterically hindrance: if the long heparin chains are not expressed, the Ads have more space to use other receptors for uptake and transduction (Table 2).

Compared with the results from the sulfation-ablated CHO-606 cells, the HSPG-ablated CHO-745 cells showed a similar trend in transduction efficiencies of analyzed Ads, supporting the fact that the sulfation status of the heparan chains play a significant role in Ad transduction.

According to literature, species D Ads can use the blood coagulation factor X (FX) as an adapter for virus uptake into the cells [36,37,38] and only some Ads from species B, C and F (type 3, -5, -35, -40 and -41) are using HSPG as receptor [9,10,11]. To study whether HSPG can be used by species D Ads for cell entry by using FX as an adapter, all following experiments focus on the influence of FX in uptake and transduction of Ad by different HSPG-depleted CHO cell lines.

### 3.3. Assay Establishment of FX effect as an Adapter for Bridging the Adenovirus to HSPG

To test the conditions for using FX as an adapter for bridging the Ad to HSPG and as proof of concept, Ad5 was used to show enhanced transduction in CHO-K1 cells, because they express the wild type form of HSPG [1,9,13,14,17,18,19,20]. The CHO-K1 cells were infected with 500 vpc of Ad5 and 10 µg/mL FX, with (1) addition of virus and FX simultaneously, with (2) incubation of the cells with FX prior for 1 h and then adding the virus and with (3) incubating FX and virus prior for 1 h and then adding it to the cells (Figure 4A). The transduction was detected by luciferase assay and is displayed in RLU (relative light units). As a control, only DMEM and virus and only OMEM and virus was used. A five-fold higher RLU compared to the controls was determined by adding virus and FX simultaneously and with preincubating FX and virus for 1 h before adding to the cells. A six-fold higher RLU was detected by incubating the CHO-K1 cells with FX prior to infection with Ad5. All following experiments were done based on the previous results with adding FX and the viruses simultaneously to simplify the procedures.

In addition, the effect of different FX concentrations was determined in a range from 6, 8 and 10 µg/mL (physiological level of FX in blood). For this purpose, CHO-K1 cells were infected with Ad5 ranging from 100 to 5000 vpc and 24 hpi, luciferase assays were performed. In Figure 4B, the results of this concentration gradient experiment are displayed. Data are shown as a ratio of treated to non-FX treated CHO-K1 cells. At 100 vpc of Ad5, the transduction efficiency is increased by FX up to 100-fold compared to non-FX treated cells and up to 160-fold for 250 vpc. Starting at 500 vpc, the effect of FX less pronounced (20-fold). However, at this dose we observed cytotoxic effects from added virus. As a conclusion, the physiological level of either 6, 8 or 10 µg/mL FX increased the transduction efficiency up to 160-fold compared to non-FX treated cells, but no significant difference can be observed when comparing only the different FX concentrations.

### 3.4. FX Has No Effect on Species D Adenovirus Transduction and Uptake

The effect of FX on all species D Ads was tested in the HSPG-depleted CHO cell lines. The CHO cells were infected with 5000 vpc (according to the data from Figure 3) of 13 species D Ads together with 6, 8 and 10 µg/mL FX. At 24 hpi, luciferase assays were carried out. Ad5 was used as a control for CHO-606 and CHO-745 cells. The data for each HSPG-depleted CHO-cell line is shown in Figure 5A–C as a ratio to non-FX treated cells. In all used CHO cell lines, the transduction of the viruses was not enhanced by using any FX concentration as adapter to bridge the viruses to HSPG.

To proove the fact that FX does not bridge the used species D Ads to HSPG, the early uptake was analyzed by duplex qPCR. As control of an increased uptake effect by FX, Ad5 was used. For this purpose, CHO-K1, -606 and -745 cells were infected with 1000 vpc of Ad5 and 10 µg/mL FX. At 3 hpi, the gDNA was isolated and duplex qPCR with GAPDH as housekeeping gene for the CHO cells and neomycin to detect the adenovirus genome was performed. In Figure 6A, the enhanced uptake effect by FX of Ad5 into CHO-K1 (blue) expressing wild type HSPG is shown as infectious particles per cell. Compared to the cells infected with Ad5 without FX, a 3-fold increased infectious particle referring to viral genome uptake were detected. For the sulfation-ablated CHO cell line 606 (gray) no significant difference between using FX or not was observed. The CHO-745 cells (orange) only keeping the proteoglycan of the HSPGs show the same result as the 606 cells but a decreased number of infectious particles were detected.

In Figure 6B, the detected Ad genome in the cells 3 hpi of each used species D Ads are shown as a ratio of treated compared to non-FX treated cells. When using FX as an adapter, no significant increase in uptake was measured in any of the used CHO cell lines. The early uptake results align with the transduction efficiency results 24 hpi, confirming no increasing effect of FX in all tested species D Ads.

## 4. Discussion

We analyzed the influence of blood-coagulation factor X (FX) on uptake and transduction of human species D Ads into host cells via HSPGs. For this purpose, three different Chinese Hamster Ovarian cell lines with depleted forms of HSPGs were used. Previous research revealed the function of FX as a bridge between adenovirus 5 (Ad5) and HSPG which functions as an entry receptor into cells with low expression levels of the main Ad5 receptor CAR. Because species D comprises the largest group of human Ads [5], and specific types such as Ad26 are explored as vaccine vectors for Ebola, Zika or SARS-Cov2 [8], this species was chosen to be further investigated on its ability for uptake and transduction by HSPG into host cells. Additionally, the role of FX as a bridge between Ad and different forms of HSPG was determined. With a physiological concentration of 6 to 10 µg/mL in blood [23,24], it is important to examine the influence of FX on Ad transduction, especially if the Ad vector is delivered intravenously. Since FX coats the hexon of Ad5 and therefore protects the Ad from being neutralized by human IgM antibodies and the classical complement pathway, to investigate the influence of FX on Ad uptake and shielding is important for Ad vectors in the context of successful in vivo delivery [14,17,25]. The influence of HSPG, heparin sulfate deficiencies and FX binding on Ad infection may also be relevant to clinical Ad infections or manifestations associated with species D Ads. HSPGs are present both on the cell surface and in the extracellular matrix throughout the body and therefore our findings may be mechanistically relevant for understanding disseminating Ad disease in immunocompromised patients.

For proof of concept, we used the well-studied Ad5 as a positive control for enhanced uptake and transduction into host cells via wild type HSPG and FX as an adapter [1,9,13,14,17,18,19,20,21,22]. The uptake of Ad5 in the presence or absence of FX showed no difference in non-sulfated HSPGs expressing CHO-606 cells similar to the results of CHO-745 cells (Figure 5B,C and Figure 6A). This observation demonstrates that the sulfation status of the HSPGs plays an important role in the uptake of Ad5 with FX as shown before in the literature [9].

All tested species D Ads from our reporter gene-tagged library [35] showed no increase in uptake and transduction with and without using FX as an adapter either in CHO-K1 cells which express the wild type form of HSPG, nor in the non-sulfated (CHO-745) or the non-expressing (CHO-606) cell lines (Figure 5 and Figure 6B). Previously, Waddington et al. 2008 [19] screened different Ad types for FX binding ability by surface plasmon resonance (SPR) experiments. Hereby, it was distinguished between strong, weak and no binding of FX. Their results demonstrated that Ad5 is a strong binder to FX, and that species D Ad17, 20 and 26 are non-binders, whereas Ad37 is a strong binder. Compared to our results, Ad5 showed enhanced uptake and transduction when using FX and for Ad17, 20 and 26, no enhanced uptake nor transduction was observed which supports the SPR results. Nevertheless, our results for species D Ad37 and FX showed no enhanced effect on uptake and transduction into any of the used CHO cell lines (Figure 5 and Figure 6B). For this reason, we conclude that Ad37:FX-complex cannot enhance uptake or transduction in the HSPG expressing CHO cell lines. A reason for this may be the missing knowledge about the Ad37:FX-complex using HSPG as receptor, because it is known that utilizing a membrane bound HSPG requires integrins as a secondary receptor for successful uptake of Ad5 into the cells [1,22]. Therefore, it may be speculated whether there is a different mechanism to enter the host cells for Ad37:FX-complex compared to Ad5:FX-complexes. Additionally, in our experiments, we used Chinese Hamster Ovarian cells which display a low expression of human integrins but instead only express some hamster integrins [39,40].

The fact that CHO cells display low human integrin expression levels [39,40] leads to the first limitation of this study. Because the uptake of Ad5 via HSPG uses integrins as a secondary receptor [1,22], more research needs to be performed in human cell lines expressing HSPG and integrins. As a promising candidate, SKOV3 cells, a human ovarian cancer cell line, can be used, because it has high HSPG expression levels [9]. On the other hand, this CHO model cell lines were used because they only express HSPG to avoid any interactions with other human Ad receptors. In addition, our used CHO models express different depleted forms of HSPG with which it is best to study the influence of sulfation and heparan chains on uptake and transduction of Ads with FX as an adapter. Additional studies may be performed that test more factors in blood that interact with the Ad virion when searching for suitable candidates for gene therapy and vaccination [17]. Performing competition assays and analyzing binding strength of known Ad interaction partners such as complement, IgMs or platelet factor 4 (PF4, associated with vaccine-induced immune thrombotic thrombocytopenia) [41,42] may provide insights into the fate of the vector after in vivo administration. Besides the influence of blood proteins, Ad interaction with blood cells such as dendritic cells, macrophages, erythrocytes, lymphocytes and granulocytes or for instance muscle cells after intramuscular application of the vector may be an important determinant about the effectiveness of the vector. 

As a conclusion, the tested species D Ads in combination with FX showed no significant enhancement in uptake and transduction via HSPG into the used CHO cell lines.

## Figures and Tables

**Figure 1 viruses-15-00055-f001:**
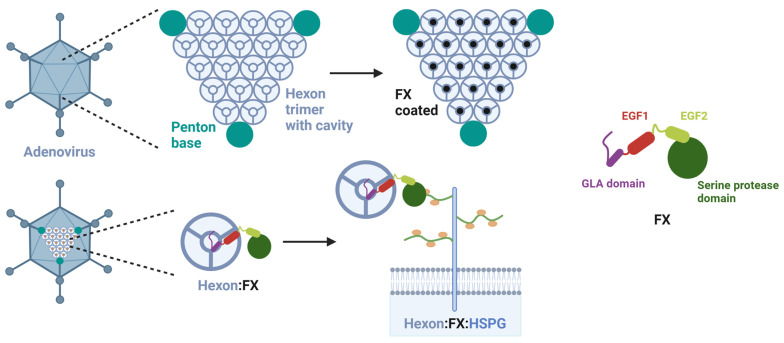
Bridging of adenovirus capsid to HSPG on host cells by FX. The 20 faces of the Ad capsid comprise 240 hexon trimers (gray). In the trimer cavity, the FX binds with its Gla domain (purple) resulting in a protruding serine protease domain (dark green). This serine protease domain bridges the Ad capsid to HSPGs on the host cells, which leads to the uptake of the Ad into the cell [19]. Figure created with Biorender.com.

**Figure 2 viruses-15-00055-f002:**
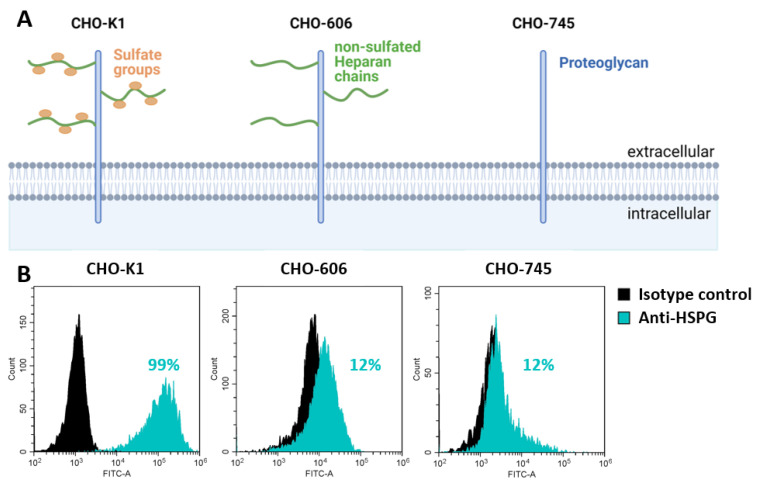
Different HSPG-expressions of CHO cell lines detected by flow cytometry. (**A**). The CHO-K1 cell line expresses the wild type form of HSPG with sulfated heparan chains on proteoglycan. CHO-606 cells have non-sulfated heparan chains and the CHO-745 cell line only the proteoglycan [10]. (**B**). The HSPG was detected by mouse anti-human heparan sulfate IgM (10E4 epitope, clone 8.S.087) and goat anti-IgM to mouse with AlexaFluor® 488 antibodies. Stated in cyan is the percentage of HSPG-positive cells inside the whole cell population. Shown are the results of two independent experiments. Further details of the determination of the FACS results are mentioned in the Appendix A.

**Figure 3 viruses-15-00055-f003:**
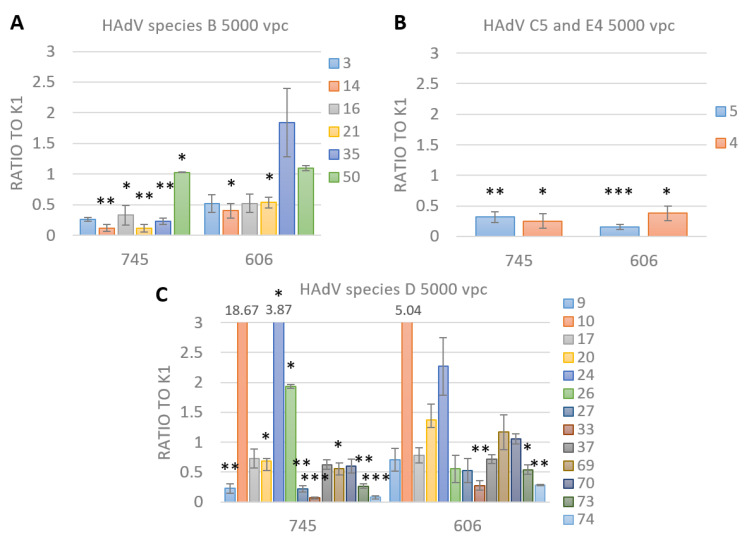
GLN-tagged adenovirus library screening in HSPG-depleted CHO cell lines. 24 h post-infection with 5,000 vpc luminescence was measured. All RLU (relative light units) from sulfation-ablated CHO-606 and heparan chain-ablated CHO-745 cells were normalized to the CHO-K1 cells with the wild type version of HSPG. For each Ad type, the fold increase compared to results obtained in CHO-K1 cells is displayed. (**A**). Species B Ads show lower or similar transduction efficiency in CHO-606 and -745 cells compared with CHO-K1 cells. (**B**). Species C and E Ads show a significant decrease in transduction efficiency compared to the wild type form of HSPG. (**C**). For species D Ads 10 out of 13 showed a decrease in transduction efficiency in CHO-606 and -745 cells compared to CHO-K1. Mean ± SEM (standard error of the mean) of n = 3 biological replicates is shown. * *p* ≤ 0.05, ** *p* ≤ 0.01, *** *p* ≤ 0.001.

**Figure 4 viruses-15-00055-f004:**
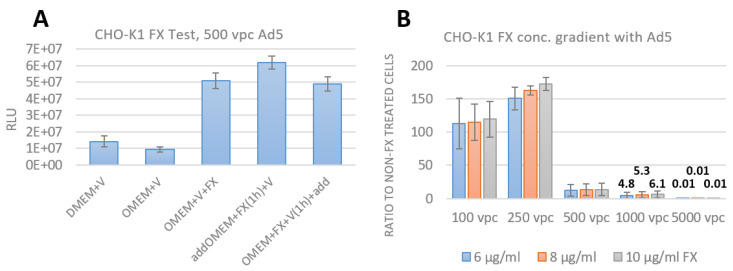
Establishment of the experimental setup for FX experiments in CHO-K1 cells. (**A**). Infection of CHO-K1 cells with 500 vpc of Ad5 and 10 µg/mL FX. As controls, virus only in DMEM and OMEM was used. As a first condition, FX and virus were added simultaneously to the cells; as the second condition, FX was preincubated for 1 h with the cells and then virus was added; as a third condition, FX and virus was preincubated for 1 h and then added to the cells. Luciferase assays were performed 24 hpi and the data are displayed in relative light units (RLU). n=2 biological replicates. (**B**). Infection of CHO-K1 cells with different vpc of Ad5 and the physiological level of FX in blood from 6 (blue), 8 (orange) to 10 µg/mL (gray). The use of 100 vpc of Ad5 with all used FX concentrations increased the transduction up to 100-fold compared to non-FX treated cells; 250 vpc up to 160-fold. Starting at 500 vpc, the increase was less pronounced (20-fold). However, at this dose, cytotoxic effects from the added virus was observed. The data are displayed as a ratio to non-FX treated cells. Mean ± SEM of n = 3 biological replicates is shown.

**Figure 5 viruses-15-00055-f005:**
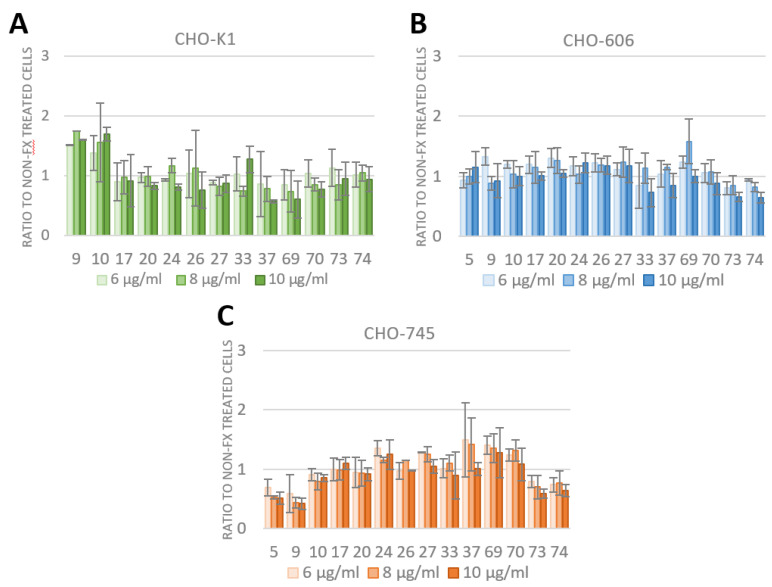
Transduction of species D adenovirus into CHO-K1, -606 and -745 cells in the presence of physiological FX concentration ranging from 6 to 10 µg/mL analyzed with luciferase assay. The CHO cells were infected with 5000 vpc of 13 different species of D Ads and either 6, 8 or 10 µg/mL FX. After 24 hpi, luciferase assays were conducted. Ad5 was used as control for CHO-606 and CHO-745 cells. The data are normalized to non-FX treated cells. There was no transduction enhancing effect of FX. (**A**). Transduction efficiencies in CHO-K1 cells. (**B**). Transduction efficiencies in CHO-606 cells. (**C**). Transduction efficiencies in CHO-745 cells. Mean ± SEM of n = 3 biological replicates.

**Figure 6 viruses-15-00055-f006:**
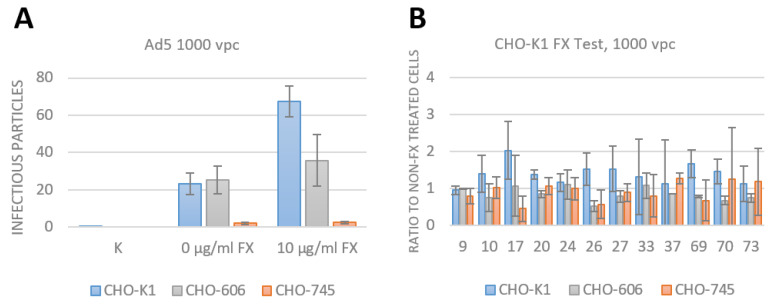
Early uptake of adenovirus 5 as control and 13 species D adenovirus into CHO cells. The cells were infected with 1000 vpc of each virus with or without 10 µg/mL FX. At 3 hpi, gDNA was isolated and duplex qPCR detecting GAPDH as housekeeping gene and neomycin contained in the Ad genome was performed. The infectious particles in the different cell lines are displayed in blue (CHO-K1 cells), grey (CHO-606 cells) and orange (CHO-745 cells). (**A**). As proof of concept, the CHO cells were infected with 1000 vpc of adenovirus 5 with and without FX. The early uptake of viral particles was 3-fold higher with FX than without in CHO-K1 cells. The number of infectious particles stayed the same in CHO-606 cells with and without FX, and in the CHO-745 cells with no heparan chains, the uptake of particles was very low under both conditions. N = 2 biological replicates. (**B**). The CHO cell lines were infected with 1000 vpc of each species D adenovirus. After 3 hpi no effect of FX was seen either in wild type HSPG CHO cells or in the HSPG depleted cells. Mean ± SEM of n = 3 biological replicates is shown.

**Table 1 viruses-15-00055-t001:** Sequence of primers and probes used in duplex qPCR for analyzing virus entry.

Primers and Probes	Sequence 5′-3′
Neo fwd	TGCTCCTGCCGAGAAAGTAT
Neo rev	GCTCTTCGTCCAGATCATC
Neo-probe-FAM	TACTCGGATGGAAGCCGGTCTTGTC
CHO GAPDH fwd	AAGGCCATCACCATCTTCCA
CHO GAPDH rev	GCGGAGATGATGACCCTCTT
CHO GAPDH probe-HEX	CTGGCGCCGAGTATGTTGTGGAATC

**Table 2 viruses-15-00055-t002:** Summary of the results from the adenovirus screening in HSPG-depleted CHO cell lines. In green are Ad types with the highest uptake detected by luciferase assay in CHO-K1 cells, in blue are Ad types with the highest uptake in -606 and in orange are Ad types with the highest uptake in -745 cells are marked. Moreover, the known receptor and adapter usage is displayed for each Ad species adapted by Arnberg et al. 2012 [36], Hensen et al. 2020 [37] and Gao et al. 2020 [38].

Adenovirus Species	Type	Receptor	Adapter
B	3, **14**, **16**, **21**, **35**, 50	CD46 ^1^, DSG2 ^2^, CD80 ^1^, CD86 ^1^	FX
C	5	CAR, HSPG, VCAM-1 ^3^, MHC1-a2 ^4^, SR ^5^	FIX ^7^, FX, Lf ^8^, DPPC ^9^
D	**9**, **10**, 17, 20, **24**, 26, **27**, **33**, **37**, 69, 70, **73**, **74**	CAR, CD46, SA ^6^	FX
E	** 4 **	CAR	

^1^ CD: cluster of differentiation; ^2^ DSG2: Desmoglein-2; ^3^ VCAM-1: Vascular cell adhesion molecule-1; ^4^ MHC1-a2: Major Histocompatibility Complex-a2; ^5^ SR: Scavenger Receptor; ^6^ SA: Sialic Acid; ^7^ FIX: Blood-coagulation factor IX; ^8^ Lf: Lactoferrin; ^9^ DPPC: Dipalmitoyl Phosphatidylcholine.

## Data Availability

Not applicable.

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
