# Peer review of "Influence of Heparan Sulfate Proteoglycans and Factor X on species D Human Adenovirus Uptake and Transduction"

_viruses, 2022, doi:10.3390/v15010055_

Round 1

Reviewer 1 Report

 The authors show in CHO cell lines that have various forms of heparin sulfate proteoglycans that sulfating plays an important role in FX-bound Ad5, but neither sulfating nor FX influenced uptake of all  species D Ads test. The results are clear and support the conclusions.

Because I'm not sure how much copyediting Viruses does, I just edited as I read and reviewed the ms. Those edits are included in the attachment. 

The authors should be sure to address the following key points (that are also noted on the edited ms.)

Lines 72-74. The sentence was long and confusing. 

Line 221. I believe the data show the opposite. I think it should say that the ...increasing transduction effect from Ad20 in CHO-6060 cells was not seen in CHO-745 cells."  (and "abolished" is too strong)

Lines 295 and 306. If duplex qPCR was done, what were the two genes?

Lines 323-325. This is about the third or fourth time you have described the 3 CHO cell lines. It should be streamlined.

Line 336. The finding  is not described ("this finding is important for Ad vectors...")

Lines 345-353 - comparing your data with previous data. This is not clearly written. I tried to edit it, but I may not have captured what you were trying to say. Please check and rewrite as necessary.

References should all be formatted the same way (probably 'sentence case' for the titles).

In the abstract the last line mentions "implications for species D Ads used as vaccine and gene therapy vectors."  The authors should discuss this in the Discussion, with some indication of what some of the implications are.

Author Response

Response to reviewer 1:

We are very grateful to this reviewer for the positive statement and the constructive suggestions, which helped to significantly improve our manuscript. Please find the answers to the specific points below.

Lines 72-74: We agree with the reviewers comment that this sentence was too long and confusing. Therefore, we split this sentences into two sentences.

Line 221: The reviewer stated that the data show the opposite as we described in the text and that the word “abolished” is too strong. Thank you for this comment. We changed the description of the data accordingly and use the word “reduced” instead of “abolished”. We hope that this clarifies this issue.

Line 295 and 306: The reviewer asked about the two genes used for the duplex qPCR. To address this comment we added the two genes detected by the duplex qPCR in both lines of the manuscript. GAPDH was used as housekeeping gene for the CHO cells and neomycin to detect the adenovirus genome.

Line 323-325: The reviewer stated that the three CHO cell lines were described for the third or fourth time and that it therefore should be streamlined. We agree with this comment and deleted the repetitive description of the CHO cell lines.

Line 336: The reviewer mentioned that “the finding” was not described properly in this line. We agree and changed that phrase to “…, to investigate the influence of FX on Ad uptake…”.

Line 345-353: The reviewer stated that the comparison with previous data is not clearly written and needs to be rewritten. We rewrote this sentences and hope that this clarifies this point.

The reviewer commented that the references should all be formatted in the same way. We changed the reference style to MDPI_7.1_References.

The reviewer suggested naming some implications of using species D Ads as vaccine and gene therapy vectors in the discussion section since this we only stated in the abstract. To address this comment we added the following section to the discussion part:

“Performing competition assays and analyzing binding strength of known Ad interaction partners such as complement, IgMs or platelet factor 4 (PF4, associated with vaccine-induced immune thrombotic thrombocytopenia) [41; 42] may provide insights into the fate of the vector after in vivo administration. Besides the influence of blood proteins, Ad interaction with blood cells such as dendritic cells, macrophages, erythrocytes, lymphocytes and granulocytes or for instance muscle cells after intramuscular application of the vector may be an important determinant about the effectiveness of the vector.”

Reviewer 2 Report

A brief summary

The manuscript compares the transduction efficiency of various adenoviruses in CHO cell lines carrying deficiencies in heparan sulfate biosynthesis in the presence or absence of human coagulation factor X. The paper aims to explore whether FX can bridge human species D Adenoviruses to cellular HSPG and augment viral transduction. The authors conclude that species D adenoviruses do not have

enhanced transduction in the presence of FX.

The strength of the paper is the adenovirus library that the authors used, which includes various adenoviruses. 

General concept comments

The authors used cell lines with heparan sulfate deficiencies and did observe the difference in transduction efficiencies. However, they did not discuss why it is important or mechanistically relevant to any clinical adenovirus infection or manifestations.

Specific comments 

Line 70. 205 FX molecules bound to viral particles, while in figure 1, all of the hexon trimers are occupied, which would result in 240 FX molecules per viral particle.

Figure 2. The isotype control has various staining intensities between cell lines. Why is that? Also needs to be clarified how the 12% was determined. Additionally, what does 12% in CHO-606 mean? Only 12% of cells have HSPG, and all of it is non-sulfated, or 12% of cells have normal HSPG, and the rest is non-sulfated? The same questions can be raised for CHO-745 data. Also, these data are not discussed in the rest of the manuscript. What is the importance of these results?

Figure 3. The comparison sample should be included in the graph with clear markings which pairs of samples were compared.

Figure 4A (and supplementary figures) need precise axis unit measurements; for example, the first two samples could be anywhere from 0 to 2e7. Therefore, the scale should be changed, or the exact numbers should be added to each bar. The exact values also should be added to Figure 4 B 1000 vpc and 5000 vpc. 

Line 270. That is a very questionable statement.

Figure 5A. What is the purpose of including Ad5 as control here?

Author Response

Response to reviewer 2:

We appreciate the comments and issues raised by this reviewer and are thankful for the suggestions that helped us to improve our manuscript. Please find the answers to these comments below.

As general comment, the reviewer mentions that we used cell lines with heparan sulfate deficiencies and did observe the difference in transduction efficiencies. However, the reviewer also points out that we should discuss why it is important or mechanistically relevant to any clinical adenovirus infection or manifestations. To address this comment we added the following paragraph to the discussion section:

“The influence of HSPG, heparin sulfate deficiencies and FX binding on Ad infection may also be relevant to clinical Ad infections or manifestations associated with species D Ads. HSPGs are present both on the cell surface and in the extracellular matrix throughout the body and therefore our findings may be mechanistically relevant for understanding disseminating Ad disease in immunocompromised patients.“

Line 70: The reviewer stated that we mention that 205 FX molecules are bound per viral particle, while in figure 1, all of the hexon trimers are occupied, which would result in 240 FX molecules per viral particle. This number was based on surface plasmon resonance (SPR) experiments mentioned in the cited reference (Waddington et al. 2008, Cell). In this reference, it is stated: “Moreover, SPR analysis indicates a stoichiometry of binding of 205 FX molecules per virus particle consistent with one FX molecule binding to each hexon trimer.” Additionally, our schematically figure 1 was adopted from the Cryoelectron Microscopy pictures in figure 3 of this reference, which is now stated in the revised version of the manuscript.

Figure 2: The reviewer is wondering why the isotype control has various staining intensities between cell lines. Furthermore, the reviewer asks for clarifying how the 12% for the CHO-745 and -606 cells were determined. Additionally, the reviewer asked what exactly 12% for CHO-606 and CHO-745 cells mean.

To address this comment we would like to first point out that we independently repeated the FACS experiments twice with all three cell lines and both times the same staining intensities were shown.  To clarify the calculated numbers for this reviewer and the reader, we included an additional section on how the anti-HSPG antibody works and how we examined the FACS data in the supplementary materials section as Method M1. We hope that this clarifies this issue.

Moreover, the reviewer states that these data are not discussed in the rest of the manuscript and wonders what the importance of these results are. As stated in the manuscript, these data were only supposed to show expression levels of HSPG in the used CHO cell lines. Potentially we could have also added this data set to the supplementary material section, but we thought it is important to further characterize and confirm features of these cell lines and to show that the sulfated HSPG specific antibody can discriminate between sulfated and non-sulfated HSPGs.

Figure 3: The reviewer suggested to include the comparison sample in the graph with clear markings which pairs of samples were compared. We decided to not include a comparison sample because the direct comparison would be the results obtained with CHO-K1 cells, which  were normalized to 1. For each Ad type the fold increase compared to results obtained in CHO-K1 cells is displayed which is now also mentioned in the figure legend. This would have meant that each graph would additionally show for each adenovirus type one extra column with a ratio to K1 of 1 which in our opinion would overload the graph and its statement.

Figure 4: The reviewer asked to change the scale of the graph or to display the exact numbers in subfigure 4A and to add the exact numbers of infectious particles at 1000 and 5000 vpc to subfigure 4B. We agree with this suggestion of this reviewer. To address this comment we changed the y-axis in subfigure 4A and the exact numbers of infectious particles was added when using 1000 and 5000 vpc in subfigure 4B.

Additionally, the reviewer asked the same for the supplementary figure S1, therefore we changed the scale of the y-axis for each graph too.

Line 270: The reviewer mentions that our statement: “Starting at 500 vpc, the effect of FX was only 20-fold higher, because of cytotoxic effects from added virus, therefore not expressing luciferase for proper detection.” is very questionable. To address this comment we clearly state in the revised version of the manuscript that starting at this dose cytotoxic effects were observed under the microscope (not shown). Please also note that these cytotoxic effects were visible for each repeat of the experiment.

Figure 5A: The reviewer is questioning the purpose of Ad5 as control in this figure. We agree with this reviewer that it is not necessary to include Ad5 in this figure. It was already shown in the figure before, so we deleted it and rewrote the figure legend and the text accordingly. We hope that this clarifies this comment.